# SiMAE: Subject-identity Separation Latent Masked Autoencoder for Multi-contrast MRI Synthesis and Uncertainty Estimation

## Abstract

Multi-contrast magnetic resonance imaging (MRI) provides complementary anatomical and pathological information, yet certain contrasts are often missing due to scan time, motion artifacts, or protocol variability. We present SiMAE, a masked autoencoder (MAE) operating in latent space that synthesizes arbitrary missing contrasts. MAE naturally fits conditional synthesis by reconstructing masked content from visible context in a single-pass, while latent space training enables semantic reconstruction, suppresses pixel space grid artifacts, and is computationally efficient. SiMAE employs a multi-contrast tokenizer with a shared encoder that maps each contrast into a common latent space and a joint decoder that outputs all contrasts simultaneously by aggregating cross-contrast cues. We train latent MAE with a two-phase curriculum: (i) pre-training with random token masking to learn general anatomical context, and (ii) fine-tuning with random contrast masking to specialize the model for missing-contrast synthesis. We introduce a subject token, regularized by a subject-identity separation (SIS) loss, that serves as a compact representation capturing anatomical identity and subject-specific features. The subject token is withheld from the decoder to impose an information bottleneck that encourages context-driven, token-level reconstruction. We further estimate uncertainty by repeatedly masking tokens and resynthesizing to generate uncertainty maps that highlight low-confidence regions. On BraTS 2021 and ADNI datasets, SiMAE achieves state-of-the-art synthesis quality and preserves fine anatomy and pathology.

## 1 Introduction

Multi-contrast magnetic resonance imaging (MRI) is widely used in both clinical practice and research as different imaging contrasts (e.g., T1-w, T2-w, FLAIR) accentuate distinct tissue properties and pathologies, providing complementary views of the same anatomy (Wu et al., 2010; Dickinson et al., 2013). Integrating these perspectives enables a more comprehensive understanding of a patient's condition (Zhang et al., 2021; Rao et al., 2022; Byeon et al., 2025). However, in real-world practice, not all MR sequences are acquired for every patient (Kronberg et al., 2022). Some contrasts may be missing due to long scan times, patient motion or discomfort, or variations in imaging protocols across institutions (Hollingsworth, 2015; Zaitsev et al., 2015; Lustig et al., 2007). Such missing modalities complicate radiological assessment and can degrade the performance of algorithms that expect complete inputs (Chan et al., 2020).

A practical remedy is missing-modality imputation, i.e., synthesizing absent contrasts from available ones. Early work explored one-to-one translation (Dar et al., 2019; Yu et al., 2019), then many-to-one fusion for a designated target (Yurt et al., 2021; Jiang et al., 2023), and more recently many-to-many synthesis that reconstructs multiple contrasts from arbitrary inputs (Chartsias et al., 2017; Sharma & Hamarneh, 2019; Meng et al., 2024). Generative adversarial networks (GANs) (Goodfellow et al., 2014) and diffusion models (Song et al., 2020; Ho et al., 2020) have been widely adopted, yet each method presents some limitations. GAN-based methods often train unstably and can exhibit mode collapse, which degrades anatomical fidelity (Bau et al., 2019; Dhariwal & Nichol, 2021). Diffusion-based methods deliver high fidelity but require many denoising iterations per image, resulting in slow inference. Moreover, maintaining strict anatomical consistency across modal-

Figure 1: **Multi-contrast MRI synthesis with missing contrasts.** We compare Ground Truth (GT) and outputs from MM-GAN (Sharma & Hamarneh, 2019) (GAN), APT (Shin et al., 2025) (Diffusion), and our SiMAE. Arrows indicate regions with differences. Per-scan inference time is measured on an A100 GPU.

ities while integrating their diverse characteristics remains challenging for generative models (Jiang et al., 2023).

In this study, we introduce SiMAE, a masked autoencoder (He et al., 2022) (MAE) that operates in latent space for multi-contrast MRI synthesis. MAE predicts masked content from visible context, which aligns with our goal of synthesizing a missing MRI contrast from the available contrasts. By moving reconstruction to latent space, the task shifts from pixel inpainting to semantic completion of latent tokens, improving efficiency and suppressing the grid artifacts seen with pixel-based MAE (Appx. 8). Compared to diffusion-based models, which require many iterative steps, SiMAE produces a result in a single-pass, making it much faster for inference while maintaining stable training. As shown in Fig. 1, SiMAE is 150× faster than APT, which is a diffusion-based model.

The multi-contrast tokenizer maps each input contrast into a shared latent space, and the model aggregates cross-contrast cues to simultaneously generate all contrasts. We train the latent MAE in two-phase curriculum. Curriculum training first learns general anatomy and then specializes to missing-contrast synthesis. In phase 1, we use random token masking to build contextual understanding, and in phase 2, we use random contrast masking so the model learns to recover absent contrasts from the available ones. We further introduce a subject token and regularize it with a subject-identity separation (SIS) loss so that subject tokens from different subjects are well separated in the embedding space, yielding a subject representation and improving the encoder's representational quality. We prepend a subject token for identity modeling and deliberately withhold it from the latent decoder. This imposes an information bottleneck that separates identity summarization from reconstruction, guiding the decoder to rely on visible token context for detail recovery. Finally, we estimate uncertainty by measuring how much the outputs change when latent tokens are repeatedly masked and resynthesized, producing maps that highlight low-confidence regions without auxiliary networks or ensembles. On BraTS 2021 and ADNI datasets, the proposed method achieves state-of-the-art PSNR and SSIM, preserves fine anatomy and pathology, and provides uncertainty maps that support clinical interpretability. Overall, our contributions are as follows:

- We propose SiMAE, a latent masked autoencoder equipped with a multi-contrast tokenizer, enabling high-quality semantic reconstruction, effective artifact suppression, and improved computational efficiency.
- We introduce a subject token, regularized by subject-identity separation (SIS) loss, to form a compact subject representation. The token is withheld from the decoder, enforcing an information bottleneck that promotes context-driven, token-level reconstruction.
- We produce uncertainty maps of synthesized output via iterative latent masking and resynthesis, highlighting low-confidence regions without extra networks or ensembles.
- We demonstrate state-of-the-art performance of SiMAE on multi-contrast MRI datasets (BraTS and ADNI), with both quantitative gains and qualitative fidelity, supported by uncertainty estimation results and comprehensive ablation studies.

## 2 RELATED WORKS

**Multi-Contrast MRI Synthesis** Missing-modality synthesis has evolved from one-to-one translation (Dar et al., 2019; Yu et al., 2019) to many-to-one translation (Yurt et al., 2021; Jiang et al., 2023) and, more recently, many-to-many models that handle arbitrary input–output sets (Chartsias et al.,

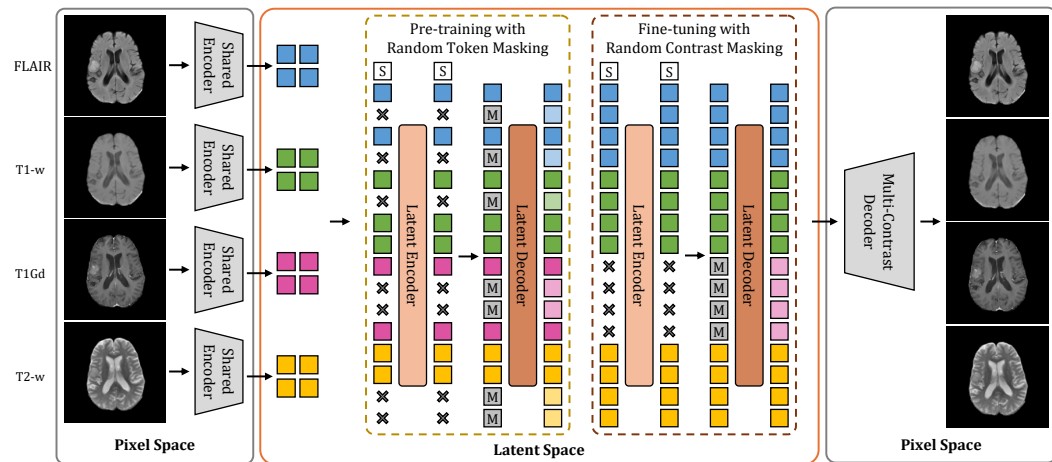

Figure 2: **Overview of SiMAE framework.** SiMAE consists of a shared pixel encoder that tokenizes all contrast images in latent space, latent encoder and decoder for reconstruct tokens in latent space, and a multi-contrast pixel decoder that transforms tokens to all contrasts simultaneously. Training follows a two-phase curriculum with random token masking and random contrast masking. At inference, when one or more contrasts are missing, SiMAE directly synthesizes the complete set of contrasts in a single-pass.

2017; Sharma & Hamarneh, 2019; Meng et al., 2024). GANs have been widely used across these settings (Goodfellow et al., 2014), including MM-GAN (Sharma & Hamarneh, 2019), ResViT (Dalmaz et al., 2022), MMT (Liu et al., 2023), and HF-GAN (Cho et al., 2024), but adversarial training can be unstable and prone to mode collapse, harming anatomical fidelity (Bau et al., 2019; Dhariwal & Nichol, 2021). Diffusion models mitigate instability and deliver strong fidelity (Song et al., 2020; Ho et al., 2020; Kazerouni et al., 2022); recent variants add joint synthesis with modality masks (Meng et al., 2024), mutual learning (Dayarathna et al., 2025), frequency-guided schedules (Xiao et al., 2024), or anatomy-aware priors (Shin et al., 2025). Nevertheless, diffusion requires many denoising steps per image, slowing inference. Our approach instead performs single-pass inference in a latent space while aggregating cross-contrast cues within one model.

**Self-supervised Learning** Self-supervised Learning (SSL) is a dominant paradigm for learning feature representations from large-scale unlabeled data (Gui et al., 2024). Masked image modeling (MIM) learns by predicting masked content from visible context (Bao et al., 2021; Xie et al., 2022); MAE (He et al., 2022) with a ViT backbone (Dosovitskiy et al., 2020) is a strong instance typically used for pre-training encoders. We repurpose masked modeling for generation by doing masking and recovery in a contrast-integrated latent space, reframing pixel-patch inpainting as semantic completion of latent tokens. Contrastive learning is another major SSL method (Chopra et al., 2005; He et al., 2020; Chen et al., 2020; Oord et al., 2018), including negative-free variants (Chen & He, 2021; Grill et al., 2020). A positive-free regularizer, dispersive loss (Wang & He, 2025), keeps only the repulsion term and has improved generative models such as DiT and SiT (Peebles & Xie, 2023; Ma et al., 2024). SIS follows the same dispersion-only principle but targets only the subject token to enforce subject-wise separation, rather than promoting generic feature diversification.

**Uncertainty Estimation in Medical Image Synthesis** For medical image synthesis to be clinically trustworthy, it should not only generate realistic images but also provide a reliable estimate of its own confidence, as deep networks can hallucinate or remove pathology (Xie et al., 2012; Ben Yedder et al., 2021). In MRI reconstruction, Bayesian inference (Narnhofer et al., 2021), Monte Carlo sampling (Edupuganti et al., 2020), and model ensembles (Zhou, 2025) are common approaches. For multi-contrast synthesis, uncertainty estimation remains underexplored: MU-Diff (Dayarathna et al., 2025) uses attention masks to indicate confidence, but there is no analysis. Inspired by MAE-DAY (Schwartz et al., 2024), we quantify output variability by repeatedly masking latent tokens of the target contrast, resynthesizing, and aggregating the differences into uncertainty maps, requiring no auxiliary networks or ensembles.

## 3 Multi-contrast MRI Synthesis

Our framework, SiMAE, operates in a latent space with continuous-valued tokens, similarly to MAR (Li et al., 2024). A multi-contrast tokenizer uses a shared encoder to map each contrast to per-contrast latent tokens in a common latent space. A joint decoder attends over the concatenated latent tokens to synthesize all contrasts simultaneously. On these tokens, SiMAE learns to complete missing content from visible context. In addition, we introduce a subject token, withhold it from the decoder, and regularize this token with a SIS loss. This design enables direct many-to-many synthesis and supports uncertainty estimation by repeatedly masking latent tokens and quantifying variability across resynthesized outputs. An overview of the training framework is shown in Fig. 2.

### 3.1 Tokenizer

We train a multi-contrast tokenizer as a KL-regularized autoencoder following LDM (Rombach et al., 2022). Training minimizes the reconstruction loss with a KL penalty on the approximate posterior and a GAN loss (Goodfellow et al., 2014). We adopt continuous latents to avoid the quantization errors that degrade reconstruction quality (Fan et al., 2024). The only difference from a standard KL-regularized autoencoder is that the multi-contrast decoder takes the concatenated tokens as input and outputs all contrasts simultaneously. After convergence, the multi-contrast tokenizer is frozen.

### 3.2 Masked Autoencoder in Latent Space

Given latent tokens $Z$ from the tokenizer, we add positional embeddings (Vaswani et al., 2017) and process the sequence with a ViT-style Transformer (Dosovitskiy et al., 2020). Let $M$ be a binary mask over the token grid and $\bar{M} = 1 - M$. The latent encoder consumes only the visible tokens (i.e., $M \odot Z$). Then, we concatenate the encoded visible tokens with the learnable mask tokens at the masked positions for the decoder input. Our latent decoder with the same architecture as the encoder predicts the full set of tokens $\hat{Z}$. The reconstruction loss is computed only on masked positions:

$$\mathcal{L}_{\text{Recon}} \ = \ \big\| \, \bar{M} \odot (Z - \hat{Z}) \, \big\|_1. \tag{1}$$

During training, we vary the masking pattern and ratio, enabling the model to recover both local details and larger missing regions.

### 3.3 Subject Token

**Subject token for identification.** We prepend a learnable subject token to the encoder input to serve as a compact embedding of the subject's identity and anatomy-specific features, as in Fig. 2. The subject token participates in self-attention as latent tokens, thereby aggregating global information while broadcasting identity-related signals back to local tokens through attention. We experimented with using multiple subject tokens (e.g., 1, 4, 8, . . . ) and found a sweet spot at 1-4 tokens; adding many tokens tends to slightly degrade the identity representation and reconstruction quality (Appx. 13). Thus, we use a single subject token by default for simplicity and efficiency.

**Information bottleneck.** Crucially, the subject token is withheld from the decoder. This creates an information bottleneck that prevents shortcutting via a subject-identity summary and compels the model to rely only on visible token context for reconstruction. This asymmetric design places identity modeling on the encoder side while keeping the decoder context-driven. Ablations show that passing the subject token to the decoder weakens detail recovery (Sec. 5.2).

### 3.4 Objectives

**Subject-identity separation.** To encourage the model to distinguish between different subjects, we introduce a subject-identity separation (SIS) loss on the subject token. SIS encourages tokens from different subjects to occupy well-separated regions of the embedding space, thereby strengthening subject recognition and improving the encoder's representational quality. Let $s_i$ denote the encoded subject token for the $i$-th scan in a batch. We adopt a dispersion-only objective that pushes subject tokens apart:

$$\mathcal{L}_{\text{SIS}} = \log \mathbb{E}_{i,j} \left[ \exp \left( -\mathcal{D} \left( s_i, s_j \right) / \tau \right) \right], \tag{2}$$

Figure 3: **Uncertainty map estimation.** After synthesizing the missing contrast, we repeatedly mask its latent tokens and resynthesize the image $N$ times. The pixel-wise variations across these resyntheses are aggregated to produce the uncertainty map.

where $\mathcal{D}$ is a dissimilarity function and $\tau$ is a temperature hyperparameter. In our experiments we use squared $\ell_2$ distance, $\mathcal{D}\left(s_i, s_j\right) = \|s_i - s_j\|_2^2$. This loss is computed once per batch, applied directly to the subject tokens, and introduces no extra parameters. The SIS loss ensures that subject tokens become distinct for each individual, capturing unique anatomical identities. In Sec. 5.2, we verify that subject tokens form subject-wise clusters, indicating subject-identity.

**Total loss.** Given a subject with $C$ contrasts, denote $X_i = \{x_i^{(1)}, \ldots, x_i^{(C)}\}$. The shared encoder maps each $x_i^{(c)}$ to latent tokens $z_i^{(c)}$ in a common latent space. For a batch, we write $\mathcal{Z} = \{Z_i\}_{i=1}^n$ with $Z_i = [s_i; z_i^{(1)}, \ldots, z_i^{(C)}]$, where $s_i$ is the prepended subject token. The total training objective is

$$\mathcal{L}(\mathcal{Z}) = \mathbb{E}_{Z_i \in \mathcal{Z}}\left[\mathcal{L}_{\text{Recon}}\left(Z_i\right)\right] + \lambda \mathcal{L}_{\text{SIS}}(\mathcal{Z}), \tag{3}$$

where $\lambda$ is a weighting hyperparameter. Since the reconstruction loss provides alignment targets for training, the SIS term focuses on repelling subject tokens from each other.

**Curriculum training.** We adopt a two-phase training curriculum that differs only in the masking pattern while keeping the architecture fixed. In phase 1, pre-training for anatomical context uses *random token masking* across the latent tokens regardless of contrast. We mask a random subset of tokens and train the model to reconstruct only the masked tokens from the visible ones. This encourages the model to leverage surrounding tokens and cross-contrast cues, learning overall context such as global layout, relative structure, intra-/inter-contrast relationships.

Then, in phase 2, fine-tuning for contrast synthesis applies *random contrast masking*, which masks all latent tokens corresponding to one or more contrasts. During fine-tuning, we turn off the SIS loss ($\lambda = 0$). This matches the test-time scenario and strengthens the model's ability to synthesize arbitrary missing contrast from the available inputs. Despite its simplicity, the two-phase curriculum training effectively improves performance. Notably, training with both masking strategies mixed in one-phase performed worse than two-phase, highlighting the benefit of first learning general context then specializing (Sec. 5.2).

## 4  UNCERTAINTY ESTIMATION

We estimate uncertainty by perturbing the latent tokens of the target synthesized contrast and measuring output variability across resyntheses. See Fig. 3 for framework of uncertainty map. First, we synthesize an initial synthesis of the missing contrast, denoted as $S_0$, using the trained SiMAE. We then reencode $S_0$ together with the available inputs through the shared encoder to obtain latent tokens. Next, we apply $N$ random masks to the tokens corresponding to $S_0$ with masking ratio $m$ and run the synthesis pipeline to obtain a set of resynthesized images, $\{S_1, \ldots, S_N\}$. Finally, the uncertainty map $U$ is computed by taking pixel-wise absolute differences between $S_0$ and each $S_i$, filtering each difference map with a Gaussian kernel $g$ (kernel size 7, $\sigma = 1.4$), and averaging:

$$U = \frac{1}{N} \sum_{i=1}^{N} \left(|S_0 - S_i| * g\right). \tag{4}$$

The uncertainty maps highlight regions where predictions are unstable. Masking in latent space creates perturbations of the model's internal process for the target contrast. Regions that are well-constrained remain stable across resyntheses, whereas ambiguous structures (e.g., lesion rims and

Table 1: Quantitative results of the comparison study on BraTS and ADNI.

| Method | BraTS | | | | | ADNI | | | |
|---|---|---|---|---|---|---|---|---|---|
| | FLAIR | T1-w | T1Gd | T2-w | Average | T1-w | T2-w | PD | Average |
| | PSNR SSIM | PSNR SSIM | PSNR SSIM | PSNR SSIM | PSNR SSIM | PSNR SSIM | PSNR SSIM | PSNR SSIM | PSNR SSIM |
| MM-GAN | 23.35 0.882 | 25.13 0.925 | 20.40 0.869 | 24.93 0.920 | 23.45 0.899 | 20.40 0.784 | 23.61 0.820 | 24.50 0.826 | 22.84 0.810 |
| HiNet | 24.36 0.898 | 23.50 0.923 | 23.64 0.886 | 26.10 0.930 | 24.40 0.909 | 20.14 0.793 | 23.97 0.838 | 24.44 0.833 | 22.85 0.821 |
| ResViT | 23.72 0.871 | 24.13 0.914 | 23.71 0.877 | 26.30 0.928 | 24.47 0.898 | 19.62 0.768 | 23.95 0.833 | 24.65 0.827 | 22.74 0.809 |
| ADM | 21.58 0.833 | 22.79 0.887 | 23.03 0.868 | 22.84 0.855 | 22.56 0.861 | 16.53 0.659 | 21.37 0.767 | 22.31 0.795 | 20.07 0.740 |
| SynDiff | 21.98 0.846 | 23.74 0.924 | 24.49 0.856 | 25.17 0.916 | 23.85 0.886 | 20.40 0.794 | 22.83 0.813 | 24.75 0.821 | 22.66 0.809 |
| M2DN | 23.83 0.870 | 24.00 0.923 | 22.65 0.899 | 24.66 0.901 | 23.79 0.898 | 19.59 0.773 | 22.99 0.777 | 23.99 0.801 | 22.19 0.784 |
| APT | 25.52 **0.920** | 25.88 0.942 | 24.54 0.903 | 27.44 **0.951** | 25.85 0.929 | 20.82 0.804 | 24.56 0.854 | 24.48 0.849 | 23.29 0.836 |
| SiMAE | **26.49** 0.916 | **27.18** **0.944** | **29.00** **0.929** | **28.61** 0.943 | **27.82** **0.933** | **26.69** **0.814** | **28.85** **0.855** | **29.07** **0.892** | **28.20** **0.854** |

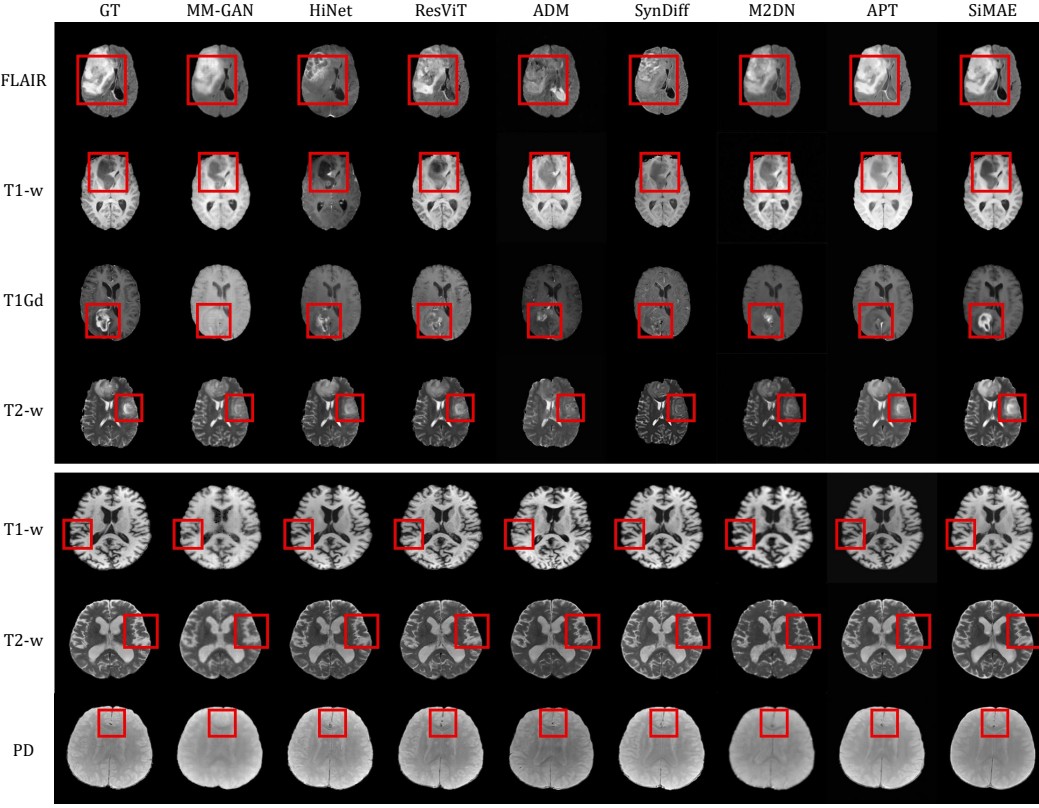

Figure 4: Qualitative results of comparison study on BraTS and ADNI. The top four rows are from BraTS, while the bottom three rows from ADNI. Major differences are highlighted by red boxes.

tissue interfaces) produce larger variability. Our approach requires no additional parameters, auxiliary networks or retraining. It can also be extended straightforwardly to scenarios with two or more missing contrasts.

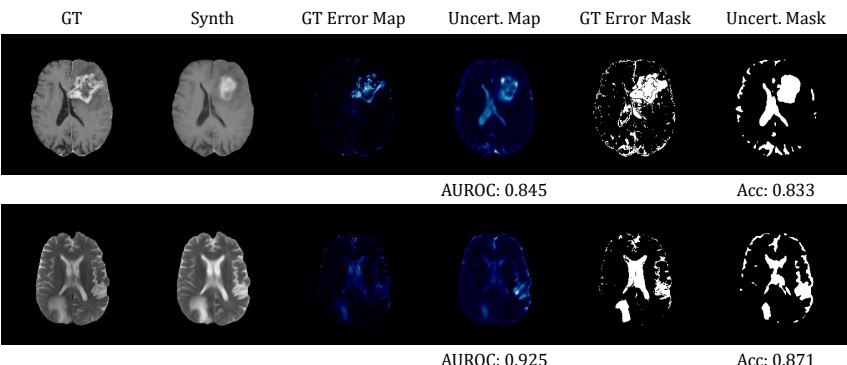

Figure 5: Uncertainty estimation: Ground Truth (GT), Synthesis, GT Error Map, Uncertainty Map, GT Error Mask (top 20%), Uncertainty Mask (top 20%).

## 5 EXPERIMENTS

**Datasets.** We evaluate on BraTS 2021 (Baid et al., 2021) and Alzheimer's Disease Neuroimaging Initiative (ADNI) datasets. BraTS comprises 2,040 multi-parametric brain tumor MRI cases with T1-weighted (T1-w), post-contrast T1-weighted (T1Gd), T2-weighted (T2-w), and T2 Fluid-Attenuated Inversion Recovery (FLAIR) volumes collected across multiple institutions, reflecting variation in imaging protocols and equipment. All images are resized to $256 \times 256$ and processed as 2D slices. We allocate 80% of the subjects for training; the test set contains 200 slices from 50 randomly selected subjects. ADNI includes 837 MRI scans from both cognitively unimpaired and Alzheimer's Disease (AD) patients. The dataset includes T1-w, T2-w, and Proton Density (PD) sequences, resized to $256 \times 256$. We use 80% of the data for training; the test set contains 150 slices from 30 randomly selected subjects.

**Evaluations.** We compared our method with seven recent methods for multi-contrast MR image synthesis: three GAN-based approaches (MM-GAN (Sharma & Hamarneh, 2019), HiNet (Zhou et al., 2020), and ResViT (Dalmaz et al., 2022)) and four diffusion-based approaches (ADM (Dhariwal & Nichol, 2021), SynDiff (Özbey et al., 2023), M2DN (Meng et al., 2024), and APT (Shin et al., 2025)). We report Peak Signal-to-Noise Ratio (PSNR) (Wang et al., 2004) and Structural Similarity (SSIM). Unless otherwise noted, all experiments compare the single-contrast missing scenarios, in which the model synthesizes one missing modality based on the available others, and report averaged results.

**Implementation details.** We use the standard ViT (Dosovitskiy et al., 2020) as our backbone. We perform pre-training for 1,000 epochs and fine-tune for 50 epochs without SIS loss. By default, the SIS loss weight $\lambda$ is 0.5, and the temperature $\tau$ is 0.5. More details can be found in Appx. B.

### 5.1 MAIN RESULTS

**Quantitative Results.** In Tab. 1, SiMAE achieves the best average performance on both datasets. On BraTS, SiMAE attains the highest PSNR for all four targets. For SSIM, it ranks first or second across all targets, with particularly strong gains on T1Gd, where both PSNR and SSIM improve by a large margin. On ADNI, SiMAE achieves the best results for every target on both metrics; PSNR gains are substantial, and SSIM improves notably on PD. Overall, SiMAE offers consistent improvements in both PSNR and SSIM across all contrasts, without modality-specific trade-offs.

**Qualitative Results.** Fig. 4 shows qualitative comparisons with other methods. In BraTS, our synthesis results better preserve fine anatomy and tumor morphology, with fewer artifacts relative to competing models. In ADNI, SiMAE recovers subtler anatomical cues, including the cortical distortions characteristic of AD. Across datasets, our method exhibits visually consistent and anatomically plausible syntheses. More results are in Appx. 10.

**Uncertainty Map.** We evaluate the reliability of the uncertainty maps by comparing them to the ground truth (GT) error. The GT error map is the absolute difference between the synthesis and the

Table 2: Random missing-modality results on BraTS, comparing SiMAE against APT.

| Input | | | | SiMAE (PSNR/SSIM) | | | | APT (PSNR/SSIM) | | | |
| --- | --- | --- | --- | --- | --- | --- | --- | --- | --- | --- | --- |
| FLAIR | T1-w | T1Gd | T2-w | FLAIR | T1-w | T1Gd | T2-w | FLAIR | T1-w | T1Gd | T2-w |
| ✓ | | | | - | **26.16/0.918** | **27.96/0.913** | **26.86/0.912** | - | 23.54/0.914 | 21.20/0.869 | 21.67/0.904 |
| | ✓ | | | **25.31/0.899** | - | **28.54/0.923** | 27.36/0.930 | 23.54/0.893 | - | 22.22/0.883 | 25.62/**0.931** |
| | | ✓ | | **24.90/0.887** | **25.67/0.921** | - | **26.81/0.913** | 22.25/0.869 | 24.27/0.912 | - | 23.24/0.898 |
| | | | ✓ | **24.91**/0.887 | **25.83/0.929** | **28.38/0.918** | - | 23.40/**0.893** | 23.93/0.925 | 23.17/0.889 | - |
| ✓ | ✓ | | | - | - | **28.83/0.926** | 28.38/0.940 | - | - | 24.26/0.899 | 27.06/**0.948** |
| ✓ | | ✓ | | - | **26.67/0.934** | - | **28.10/0.931** | - | 24.74/0.930 | - | 23.03/0.924 |
| ✓ | | | ✓ | - | **26.97/0.939** | **28.47/0.924** | - | - | 24.50/0.938 | 21.14/0.886 | - |
| | ✓ | ✓ | | **25.85/0.904** | - | - | 27.80/0.935 | 24.08/0.898 | - | - | 26.52/**0.939** |
| | ✓ | | ✓ | 26.15/0.912 | - | **28.99/0.928** | - | **26.40/0.923** | - | 23.30/0.896 | - |
| | | ✓ | ✓ | **26.17**/0.909 | 26.65/0.940 | - | - | 25.26/**0.914** | 25.11/0.935 | - | - |

GT image. Both the GT error map and the predicted uncertainty map are thresholded at their top 20% values to form binary masks. We then compute pixel-wise accuracy between these two masks. Treating the GT error mask as the reference label, we also compute the Area Under the Receiver Operating Characteristic Curve (AUROC) using uncertainty map. All metrics are computed within the brain region only.

Fig. 6 shows accuracy and AUROC versus masking ratio and number of resyntheses. First, fixing the number of resyntheses to $N = 32$, we vary the masking ratio $m$ applied to the latent tokens of the initially synthesized target $S_0$. Performance peaks at $m = 0.8$, yielding the highest mask accuracy and AUROC. Second, fixing $m = 0.8$, we vary $N$. AUROC increases gradually with $N$, but gains taper while computation grows linearly, so we set $N = 32$ for all experiments. Detailed values are in Appx. 7.

Fig. 5 visualizes our uncertainty estimation results on BraTS. High uncertainty regions are located particularly along lesion and tissue boundaries with large synthesis errors. This indicates that the uncertainty map highlights structures with low confidence that are indeed difficult to reconstruct. More results are in Appx. 9.

Figure 6: Accuracy and AUROC for the uncertainty map are plotted against masking ratio and the number of resyntheses.

### 5.2 ABLATION STUDIES

**Scenarios of Random Missing Modalities.** We further evaluate robustness when two or more contrasts are missing. In Tab. 2, SiMAE outperforms APT in most random-missing scenarios, particularly in terms of PSNR. As the number of available inputs decreases, the performance of both methods drops, but the gap in favor of SiMAE widens, indicating more effective use of cross-contrast context. These results indicate that SiMAE scales reliably from the single-missing setting to more challenging multi-missing scenarios with consistent improvements over a diffusion-based model. Results on ADNI are in Appx. 8.

**Subject Token Design and SIS Loss.** Tab. 3 shows the effect of the subject token, the information bottleneck, and SIS loss. Adding the subject token alone underperforms the baseline, indicating that when the token is available to the decoder it acts as a shortcut and weakens context modeling. Withholding the token from the decoder improves performance by creating an information bottleneck, which supports our claim that forcing the decoder to rely on visible, token-level evidence rather than summarized subject-identity is more effective. Applying SIS loss to the subject token yields enhanced results by making embeddings more distinct and thus strengthening the encoder's representation. The full design achieves the best results on both datasets, which indicates that subject-aware encoding and context-only decoding are effective.

Table 3: Ablation study on subject token (S.T.), information bottleneck (I.B.), and SIS loss.

|  | S.T. | I.B. | SIS | BraTS | | ADNI | |
| --- | --- | --- | --- | --- | --- | --- | --- |
|  |  |  |  | PSNR | SSIM | PSNR | SSIM |
| Baseline |  |  |  | 27.63 | 0.930 | 27.37 | 0.839 |
|  | ✓ |  |  | 27.12 | 0.928 | 27.26 | 0.837 |
|  | ✓ | ✓ |  | 27.70 | 0.932 | 27.67 | 0.846 |
|  | ✓ |  | ✓ | 27.59 | 0.930 | 28.05 | 0.849 |
| SiMAE | ✓ | ✓ | ✓ | **27.82** | **0.933** | **28.20** | **0.854** |

Table 4: Training strategy with the baseline model. (RTM: Random Token Masking, RCM: Random Contrast Masking)

| Strategy | BraTS | | ADNI | |
| --- | --- | --- | --- | --- |
|  | PSNR | SSIM | PSNR | SSIM |
| RTM | 26.91 | 0.927 | 27.11 | 0.838 |
| RCM | 26.34 | 0.919 | 27.09 | 0.837 |
| RTM+RCM | 25.87 | 0.915 | 26.32 | 0.819 |
| RTM→RCM | **27.63** | **0.930** | **27.37** | **0.839** |

Fig. 7 visualizes the subject tokens with t-SNE (Maaten & Hinton, 2008). For each subject, we plot four slices. With SIS loss, all cases form compact, well-separated clusters, indicating stronger subject-wise separability. Without SIS loss, some cases exhibit partial intermingling, which indicates weaker separation. This supports that SIS loss tightens within-subject cohesion while increasing between-subject separation.

**One-phase vs. Two-phase Training.** In Tab. 4, we use the baseline model without a subject token to compare training strategy. "Random token masking" applies random token-level masks throughout training. "Random contrast masking" excludes all tokens of one or more contrasts to mirror the contrast missing setting. "Random token + contrast masking" randomly selects one of the two masking for each iteration. For a fair comparison the total training epoch is kept constant across one-phase and two-phase setups. The two-phase training curriculum, which involves pre-training with random token masking followed by fine-tuning with random contrast masking, achieves the best performance. Using both masking schemes within one phase results in performance poorer than using one scheme. Moreover, random token masking outperforms random contrast masking even though the latter matches the inference configuration, suggesting that learning broad contextual relationships among tokens provides a stronger foundation for contrast-specific synthesis.

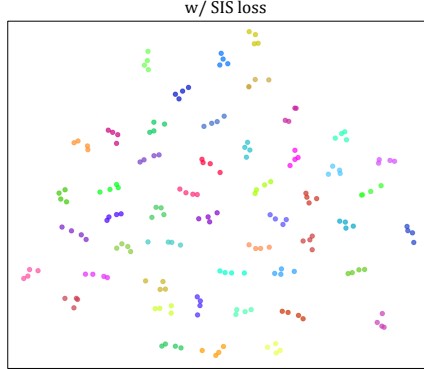

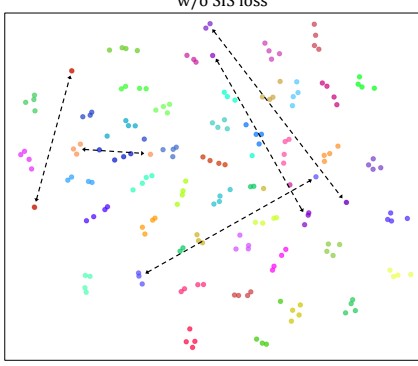

Figure 7: t-SNE visualization of subject tokens, colored by subject. Cases that fail clustering are with dashed lines.

## 6 CONCLUSION

We present SiMAE, a framework for multi-contrast MRI synthesis that combines a multi-contrast tokenizer with a latent masked autoencoder. The subject token, regularized by SIS and withheld from the decoder to impose an information bottleneck, improves encoder representations and promotes detail-faithful reconstruction. SiMAE also produces uncertainty maps via iterative latent masking that align with ground truth error and highlight low-confidence regions. Ablations show that a two-phase curriculum, consisting of random token masking for context followed by random contrast masking for specialization, outperforms one-stage training. SiMAE achieves state-of-the-art results on BraTS and ADNI. Future directions include using the uncertainty map to refine uncertain regions through resynthesis or loss reweighting. We also plan to extend the approach to full 3D and other imaging modalities, as well as mixed-modality settings.

**Limitations.** SiMAE is an assistive tool and synthesized contrasts should be interpreted with qualified expert oversight. As shown in Appx. 11, failure cases include boundary smoothing, contrast-appearance shifts, and small-lesion hallucinations. Uncertainty maps may also be incorrect, so they should be regarded as supplementary indicators rather than ground truth and used with caution.

**Ethics Statement.** This work uses publicly available, de-identified MRI datasets (BraTS 2021, ADNI) collected under institutional review and participant consent by the dataset curators.

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

## A  THE USE OF LARGE LANGUAGE MODELS (LLMS)

During the preparation of this work, the authors used OpenAI's GPT-5 in order to improve language and readability. After using this tool/service, the authors reviewed and edited the content as needed, taking full responsibility for the content of the publication.

## B  IMPLEMENTATION DETAILS.

All experiments were conducted on NVIDIA A100 GPUs. We follow LDM (Rombach et al., 2022) to train multi-contrast tokenizer to encode each MR image into $16 \times 16$ continuous tokens, and the multi-contrast decoder takes the concatenated latent tokens. SiMAE uses 10 transformer blocks with width 768 in both latent encoder and latent decoder. We train with the AdamW (Loshchilov & Hutter, 2017) (learning rate 5e-4, weight decay 0.02, $\beta_1 = 0.9$, $\beta_2 = 0.95$), batch size 128, and maintain an exponential moving average of parameters with momentum 0.9999. During pre-training, we train for 1,000 epochs and apply a 100-epoch linear lr warmup (Goyal et al., 2017), followed by a constant (Peebles & Xie, 2023) lr schedule, and we sample a masking ratio in [0.2, 0.8] for random token masking. Fine-tuning runs for 50 epochs without SIS loss. The SIS loss weight $\lambda = 0.5$ and the temperature $\tau = 0.5$ are selected via the ablation in Appx. 19.

## C  SUPPLEMENTARY EXPERIMENTAL RESULTS

Unless noted, we use the same model capacity, training budget, and data preprocessing for in all experiments.

### C.1  LATENT-BASED MAE VS. PIXEL-BASED MAE

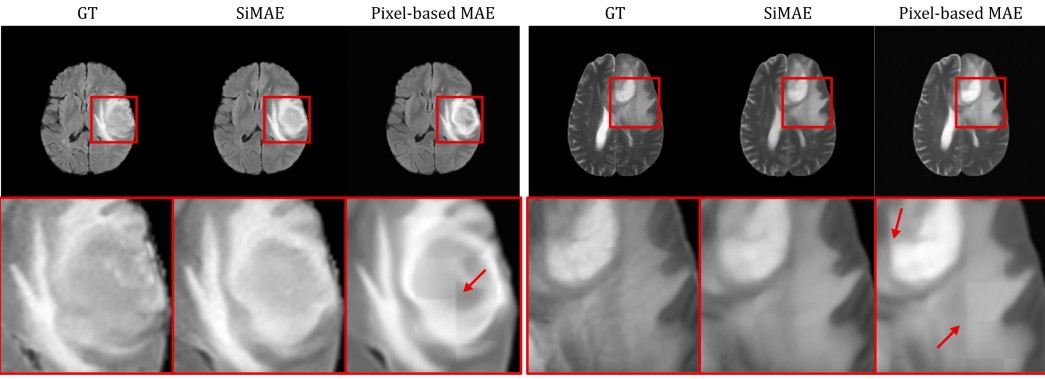

Figure 8: Qualitative comparison of MAE in latent space (SiMAE) vs. pixel space. Pixel-based MAE shows grid-like artifacts.

Fig. 8 compares qualitative results from a latent space MAE (SiMAE) and a pixel space MAE. To equalize computational cost, the pixel-based MAE patchifies each $256 \times 256$ image into $16 \times 16$ patches, and SiMAE uses a tokenizer stride that produces the same $16 \times 16$ latent tokens. We keep encoder/decoder depth and width, masking ratio, epochs, and training curriculum identical. Pixel space training consistently exhibits grid-like artifacts and boundary softening, whereas latent space training suppresses checkerboard patterns and preserves tissue interfaces.

This result is expected: pixel-based MAE is originally proposed for self-supervised representation learning rather than high-fidelity image synthesis, so patch-wise reconstruction can imprint the sampling grid. In contrast, SiMAE predicts latent tokens, and the multi-contrast decoder acts as a learned image prior that maps these tokens back to image space with spatially coherent structure. As a result, small token-level errors are smoothed in decoding, reducing artifacts. While this may not be an exact comparison, moving the reconstruction to latent space reduces grid artifacts and preserves fine anatomy more effectively.

Table 5: SIS vs. contrastive with different dissimilarity functions on BraTS.

| | FLAIR | | T1-w | | T1Gd | | T2-w | | Average | |
| --- | --- | --- | --- | --- | --- | --- | --- | --- | --- | --- |
| Method | PSNR | SSIM | PSNR | SSIM | PSNR | SSIM | PSNR | SSIM | PSNR | SSIM |
| Baseline | 26.39 | 0.911 | 26.67 | 0.940 | **29.04** | 0.928 | 28.43 | 0.940 | 27.63 | 0.930 |
| SIS ($\ell_2$) | **26.49** | **0.916** | **27.18** | **0.943** | 29.00 | **0.929** | 28.61 | **0.943** | **27.82** | **0.933** |
| Contrastive ($\ell_2$) | 26.18 | 0.910 | 26.60 | 0.938 | 28.86 | 0.927 | 28.22 | 0.937 | 27.47 | 0.928 |
| SIS (cosine) | 26.06 | 0.914 | 27.02 | 0.942 | 28.88 | **0.929** | **28.86** | **0.943** | 27.70 | 0.932 |
| Contrastive (cosine) | 25.44 | 0.907 | 26.64 | 0.939 | 28.85 | 0.927 | 28.18 | 0.937 | 27.28 | 0.928 |

Table 6: SIS vs. contrastive with different dissimilarity functions on ADNI.

| | T1-w | | T2-w | | PD | | Average | |
| --- | --- | --- | --- | --- | --- | --- | --- | --- |
| Method | PSNR | SSIM | PSNR | SSIM | PSNR | SSIM | PSNR | SSIM |
| Baseline | 25.87 | 0.796 | 28.03 | 0.840 | 28.21 | 0.881 | 27.37 | 0.839 |
| SIS ($\ell_2$) | **26.69** | **0.814** | 28.85 | **0.855** | **29.07** | **0.892** | **28.20** | **0.854** |
| Contrastive ($\ell_2$) | 25.45 | 0.786 | 27.75 | 0.835 | 28.05 | 0.879 | 27.09 | 0.833 |
| SIS (cosine) | 26.43 | 0.810 | **28.89** | **0.855** | 28.99 | 0.890 | 28.10 | 0.852 |
| Contrastive (cosine) | 25.82 | 0.794 | 27.92 | 0.838 | 28.07 | 0.879 | 27.27 | 0.837 |

## C.2 COMPARISON WITH CONTRASTIVE LOSS

We compare a contrastive loss (Oord et al., 2018) with SIS loss. Following Grill et al. (2020); Wang & Isola (2020); Wang & He (2025), we write objective

$$\mathcal{L}_{\text{Contrast}} = \mathcal{D}(z_i, z_i^+)/\tau + \log \sum_j \exp(-\mathcal{D}(z_i, z_j)/\tau), \tag{5}$$

where $(z_i, z_i^+)$ denotes a positive pair and $(z_i, z_j)$ denotes any pair of samples (positive pair & all negative pairs). We use the negative cosine similarity for dissimilarity function: $\mathcal{D}(z_i, z_j) = -\frac{z_i^\top z_j}{\|z_i\|\|z_j\|}$. In our experiments, positive pairs are constructed by varying the latent-masking pattern for the same scan.

Tab. 5 and Tab. 6 show that adding a contrastive loss degrades results relative to the baseline for both dissimilarity functions (cosine and $\ell_2$). Whereas SIS loss consistently improves performance; among dissimilarities, $\ell_2$ outperforms cosine. A contrastive loss forces the subject token from two masked views of the same scan to be nearly identical. This suppresses view-specific signal that should instead be captured by local tokens for accurate reconstruction. The reconstruction loss already provides the alignment target, so extra alignment is redundant and discards useful information. SIS keeps only a dispersion term, enlarging inter-subject distances without interference.

## C.3 UNCERTAINTY MAP

Fig. 9 illustrates uncertainty estimation results. We resynthesize the target contrast multiple times with random latent masking and aggregate the resulting differences to produce an uncertainty map. Top-20% uncertainty masks align with top-20% GT error regions, indicating that iterative latent masking captures intrinsic ambiguity of the synthesis task.

**Number of resyntheses $N$ and masking ratio $m$.** Tab. 7 shows accuracy and AUROC for different combinations of $(N, m)$ on BraTS. With $N=32$ fixed and $m$ varied, both average metrics peak at $m=0.8$. With $m=0.8$ fixed and $N$ varied, accuracy and AUROC improve up to $N=32$, after which they show only minimal improvement. As $N$ increases, computation grows accordingly, so we set $N=32$ as the default.

## C.4 ADDITIONAL QUALITATIVE RESULTS

Fig. 10 presents additional qualitative results on BraTS and ADNI. Red boxes highlight representative differences around lesion boundaries, tumor area, and the interface between each tissue. Our model achieves more accurate synthesis and better preserves anatomical details.

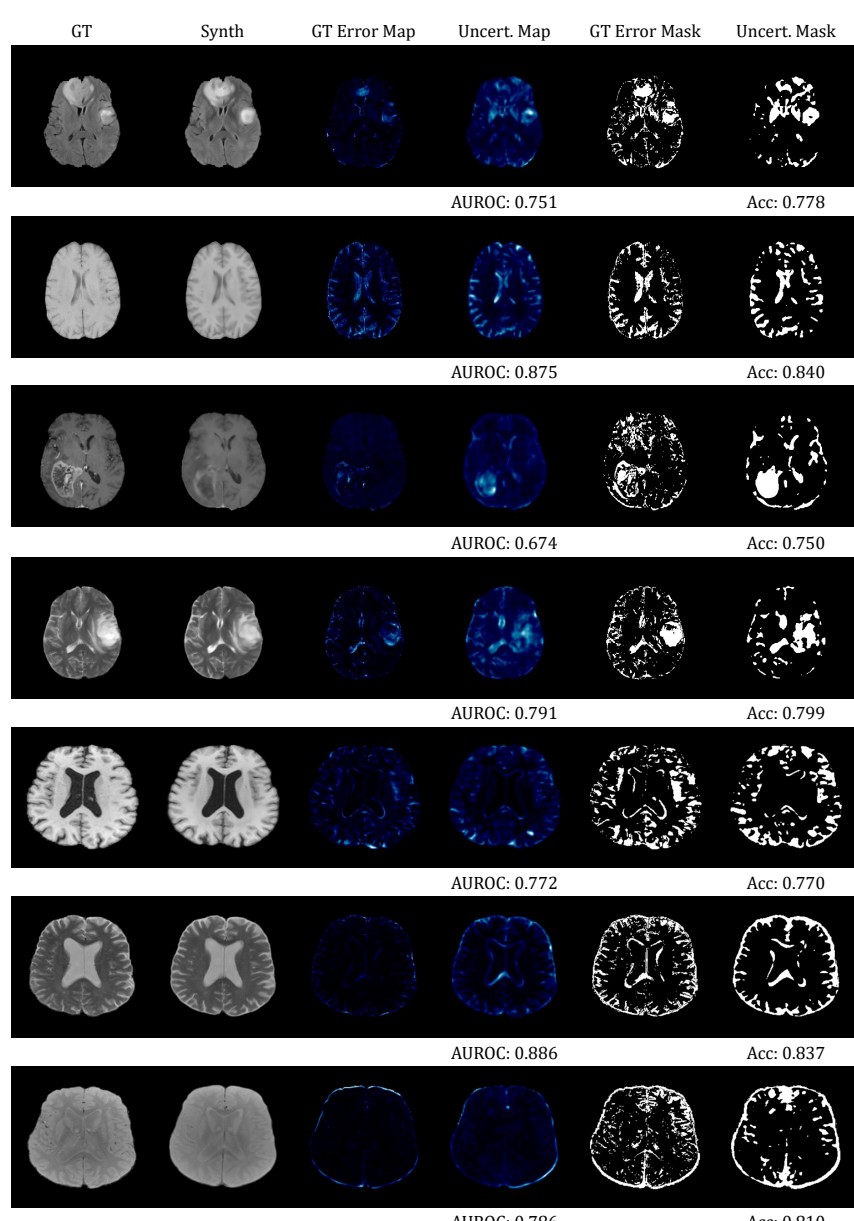

Figure 9: Uncertainty estimation: Ground Truth (GT), Synthesis, Error Map, Uncertainty Map, Error Mask (top 20%), and Uncertainty Mask (top 20%). BraTS (top four rows, FLAIR/T1-w/T1Gd/T2-w) and ADNI (bottom three rows, T1-w/T2-w/PD).

## C.5    ADDITIONAL QUANTITATIVE RESULTS

**Scenarios of Random Missing modality on ADNI.**    Tab. 8 shows results on ADNI and SiMAE outperforms APT in all random missing scenarios. Performance improvement is particularly noticeable when only PD is available.

**Curriculum training results on SiMAE.**    Tab. 9 and Tab. 10 show the results of pre-training and subsequent fine-tuning. From the pre-trained model, fine-tuning with random contrast masking improves for every contrast on BraTS and ADNI. These improvements corroborate that the fine-tuning specializes the model for missing contrast synthesis beyond the context learned during pre-training. Additionally, whether or not SIS loss is applied during fine-tuning, there is no significant

Table 7: Results of uncertainty estimation for the number of resyntheses $N$ and masking ratio $m$ on BraTS.

| | | FLAIR | | T1-w | | T1Gd | | T2-w | | Average | |
|---|---|---|---|---|---|---|---|---|---|---|---|
| $N$ | $m$ | Acc | AUROC | Acc | AUROC | Acc | AUROC | Acc | AUROC | Acc | AUROC |
| 32 | 0.01 | 0.707 | 0.579 | 0.686 | 0.511 | 0.689 | 0.539 | 0.728 | 0.646 | 0.702 | 0.569 |
| 32 | 0.1 | 0.717 | 0.608 | 0.688 | 0.519 | 0.694 | 0.540 | 0.745 | 0.692 | 0.711 | 0.590 |
| 32 | 0.2 | 0.718 | 0.611 | 0.691 | 0.530 | 0.696 | 0.541 | 0.749 | 0.697 | 0.713 | 0.595 |
| 32 | 0.3 | **0.719** | **0.611** | 0.694 | 0.538 | 0.698 | 0.544 | 0.749 | 0.698 | 0.715 | 0.598 |
| 32 | 0.4 | **0.719** | **0.611** | 0.698 | 0.547 | 0.700 | 0.546 | 0.749 | 0.697 | 0.716 | 0.600 |
| 32 | 0.5 | 0.718 | 0.610 | 0.701 | 0.555 | 0.702 | 0.548 | 0.749 | 0.697 | 0.717 | 0.603 |
| 32 | 0.6 | 0.716 | 0.606 | 0.705 | 0.565 | 0.704 | 0.551 | 0.750 | 0.697 | 0.719 | 0.605 |
| 32 | 0.7 | 0.714 | 0.600 | 0.709 | 0.574 | 0.706 | 0.552 | 0.752 | 0.699 | 0.720 | 0.606 |
| 32 | 0.8 | 0.709 | 0.588 | 0.713 | 0.585 | 0.707 | 0.550 | 0.754 | 0.704 | **0.721** | **0.607** |
| 32 | 0.9 | 0.703 | 0.570 | **0.715** | **0.594** | 0.707 | 0.548 | **0.756** | **0.707** | 0.720 | 0.605 |
| 32 | 0.99 | 0.701 | 0.553 | 0.701 | 0.563 | **0.712** | **0.562** | 0.748 | 0.691 | 0.716 | 0.592 |
| 1 | 0.8 | 0.700 | 0.550 | 0.703 | 0.555 | 0.698 | 0.534 | 0.728 | 0.625 | 0.707 | 0.566 |
| 2 | 0.8 | 0.705 | 0.567 | 0.708 | 0.576 | 0.699 | 0.536 | 0.738 | 0.657 | 0.712 | 0.584 |
| 4 | 0.8 | 0.707 | 0.576 | 0.710 | 0.578 | 0.702 | 0.544 | 0.746 | 0.677 | 0.716 | 0.594 |
| 8 | 0.8 | **0.710** | 0.589 | 0.710 | 0.578 | 0.706 | 0.549 | 0.750 | 0.690 | 0.719 | 0.602 |
| 16 | 0.8 | **0.710** | 0.588 | 0.712 | 0.582 | 0.706 | **0.551** | 0.753 | 0.698 | 0.720 | 0.605 |
| 32 | 0.8 | **0.710** | 0.588 | **0.713** | 0.585 | **0.708** | **0.551** | **0.755** | 0.704 | **0.722** | 0.607 |
| 64 | 0.8 | **0.710** | 0.591 | **0.713** | 0.585 | **0.708** | **0.551** | **0.755** | 0.706 | **0.722** | 0.608 |
| 128 | 0.8 | **0.710** | **0.593** | **0.713** | **0.586** | **0.708** | **0.551** | **0.755** | **0.707** | **0.722** | **0.609** |

Table 8: Random missing-modality results on ADNI, comparing SiMAE against a APT.

| Input | | | SiMAE (PSNR/SSIM) | | | APT (PSNR/SSIM) | | |
|---|---|---|---|---|---|---|---|---|
| T1-w | T2-w | PD | T1-w | T2-w | PD | T1-w | T2-w | PD |
| ✓ | | | - | **26.92/0.824** | **26.28/0.863** | - | 22.56/0.802 | 22.43/0.799 |
| | ✓ | | **25.56/0.804** | - | **27.87/0.890** | 20.37/0.798 | - | 23.98/0.851 |
| | | ✓ | **23.26/0.720** | **26.38/0.795** | - | 17.04/0.663 | 22.05/0.764 | - |

difference in performance. Since fine-tuning runs for fewer epochs than pre-training, its impact appears negligible. In experiments on BraTS, performance is better when sis loss is not applied, so this is set as the default.

**Ablation results on masking ratios.** Tab. 11 compares the random token masking ratios on BraTS. Performance is optimal within the 0.2–0.8 range, so this is selected as the default.

$\ell_1$ **vs.** $\ell_2$ **for reconstruction.** Tab. 12 compares $\ell_1$ and $\ell_2$ for $\mathcal{L}_{\text{Recon}}$ on BraTS. $\ell_1$ yields higher PSNR and SSIM on every contrast, so we adopt $\ell_1$.

**Number of subject tokens.** Tab. 13 varies the number of subject tokens. We find a sweet spot around 1–4 tokens and adding many tokens slightly degrades performance. This is probably because a large number of tokens fail to properly reflect the subject-identity and do not provide a beneficial effect. We set the number of subject tokens to 1 by default.

**Ablation: subject token, information bottleneck, SIS loss.** Tab. 14 and Tab.15 shows the effect of the subject token, the information bottleneck, and SIS loss. SiMAE achieves the best results on both datasets, which indicates that subject-aware encoding and context only decoding are effective.

**One-phase vs. Two-phase training.** Tab. 16 and Tab. 17 compare one-phase training-random token masking (RTM), random contrast masking (RCM), and their combination—against two-stage curriculum training (pre-training with, then fine-tuning with RCM). Curriculum training is best in all contrasts, supporting the "general context first, contrast-aware specialization later" design.

Table 9: Results of pre-training and fine-tuning on BraTS.

| | FLAIR | | T1-w | | T1Gd | | T2-w | | Average | |
|---|---|---|---|---|---|---|---|---|---|---|
| | PSNR | SSIM | PSNR | SSIM | PSNR | SSIM | PSNR | SSIM | PSNR | SSIM |
| Pre-training | 25.71 | 0.906 | 25.91 | 0.936 | 27.89 | 0.923 | 26.47 | 0.934 | 26.50 | 0.925 |
| Fine-tuning w/ SIS | 26.48 | 0.916 | 27.11 | 0.944 | 28.92 | 0.929 | 28.60 | 0.943 | 27.78 | 0.933 |
| Fine-tuning w/o SIS | **26.49** | 0.916 | **27.18** | 0.944 | **29.00** | 0.929 | **28.61** | 0.943 | **27.82** | 0.933 |

Table 10: Results of pre-training and fine-tuning on ADNI.

| | T1-w | | T2-w | | PD | | Average | |
|---|---|---|---|---|---|---|---|---|
| Method | PSNR | SSIM | PSNR | SSIM | PSNR | SSIM | PSNR | SSIM |
| Pre-training | 26.67 | 0.814 | 28.85 | 0.855 | 28.74 | 0.892 | 28.09 | 0.854 |
| Fine-tuning w/ SIS | 26.68 | 0.814 | 28.85 | 0.855 | 29.07 | 0.892 | 28.20 | 0.854 |
| Fine-tuning w/o SIS | **26.69** | 0.814 | 28.85 | 0.855 | 29.07 | 0.892 | 28.20 | 0.854 |

**Comparison with diffusion-based models.** Tab. 18 compares the number of parameters and per-scan inference time with APT (Shin et al., 2025), diffusion-based model. Time is computed with a single A100 GPU. SiMAE has a comparable number of parameters to APT, but its inference speed is over 150 times faster.

**SIS loss hyperparameters.** Tab. 19 varies the SIS weight $\lambda$ and temperature $\tau$. Performance on BraTS is stable within the range, and $(\lambda, \tau) = (0.5, 0.5)$ is slightly higher, so it is set as the default.

**Failure cases.** We show some failure cases in Fig. 11. First, the synthesized image preserves anatomy but exhibits an incorrect target-contrast appearance. Second, the uncertainty map fails to flag errors: the synthesis is acceptable but uncertainty estimation has failed. Third, tumors are partially missed and the uncertainty map also fail do not detect them, indicating that the uncertainty map can under-report failures. These cases promote expert oversight and verification, and underscore that uncertainty maps are supplementary indicators rather than ground truth.

Table 11: Results of random masking ratio on BraTS.

| Mask ratio | FLAIR PSNR | SSIM | T1-w PSNR | SSIM | T1Gd PSNR | SSIM | T2-w PSNR | SSIM | Average PSNR | SSIM |
|---|---|---|---|---|---|---|---|---|---|---|
| 0.1–0.9 | 25.57 | 0.909 | 26.53 | 0.940 | 29.07 | 0.928 | 28.77 | 0.940 | 27.49 | 0.929 |
| 0.2–0.8 | **26.39** | **0.911** | **26.67** | **0.940** | **29.04** | **0.928** | 28.43 | 0.940 | **27.63** | **0.930** |
| 0.3–0.7 | 25.76 | 0.911 | 26.51 | 0.940 | 28.86 | 0.927 | 28.42 | 0.940 | 27.39 | 0.929 |

Table 12: Reconstruction loss comparison ($\ell_1$ vs. $\ell_2$) on BraTS.

| Loss | FLAIR PSNR | SSIM | T1-w PSNR | SSIM | T1Gd PSNR | SSIM | T2-w PSNR | SSIM | Average PSNR | SSIM |
|---|---|---|---|---|---|---|---|---|---|---|
| $\ell_1$ | **26.39** | **0.911** | **26.67** | **0.940** | **29.04** | **0.928** | **28.43** | **0.940** | **27.63** | **0.930** |
| $\ell_2$ | 25.09 | 0.905 | 26.39 | 0.938 | 28.46 | 0.925 | 27.88 | 0.935 | 26.96 | 0.925 |

Table 13: Results of the number of subject tokens on BraTS.

| # of subject tokens | FLAIR PSNR | SSIM | T1-w PSNR | SSIM | T1Gd PSNR | SSIM | T2-w PSNR | SSIM | Average PSNR | SSIM |
|---|---|---|---|---|---|---|---|---|---|---|
| 1 | 26.49 | **0.916** | **27.18** | **0.944** | **29.00** | **0.929** | 28.61 | **0.943** | **27.82** | **0.933** |
| 4 | **26.63** | **0.916** | 27.01 | 0.943 | 28.87 | **0.929** | 28.77 | **0.943** | **27.82** | **0.933** |
| 8 | 26.25 | 0.912 | 26.93 | 0.941 | 28.97 | 0.928 | **28.79** | 0.942 | 27.74 | 0.931 |
| 16 | 26.26 | 0.911 | 27.02 | 0.942 | **29.00** | **0.929** | 28.60 | 0.942 | 27.72 | 0.931 |
| 32 | 26.40 | 0.915 | 26.73 | 0.941 | 28.88 | 0.928 | 28.78 | 0.943 | 27.70 | 0.932 |
| 64 | 26.55 | **0.916** | 26.58 | 0.941 | 28.86 | 0.928 | 28.51 | 0.942 | 27.63 | 0.932 |
| 128 | 26.20 | 0.913 | 26.93 | 0.941 | 28.80 | 0.927 | 28.72 | 0.941 | 27.66 | 0.931 |

Table 14: Ablation on subject token (S.T.), information bottleneck (I.B.), and SIS loss on BraTS.

| | S.T. | I.B. | SIS | FLAIR PSNR | SSIM | T1-w PSNR | SSIM | T1Gd PSNR | SSIM | T2-w PSNR | SSIM | Average PSNR | SSIM |
|---|---|---|---|---|---|---|---|---|---|---|---|---|---|
| Baseline | | | | 26.39 | 0.911 | 26.67 | 0.940 | 29.04 | 0.928 | 28.43 | 0.940 | 27.63 | 0.930 |
| | ✓ | | | 24.60 | 0.905 | 26.58 | 0.939 | 28.89 | 0.927 | 28.41 | 0.939 | 27.12 | 0.928 |
| | ✓ | ✓ | | 25.99 | 0.913 | 27.01 | 0.943 | 28.97 | **0.929** | 28.84 | **0.943** | 27.70 | 0.932 |
| | ✓ | | ✓ | 26.16 | 0.909 | 26.96 | 0.942 | 28.60 | 0.928 | **28.63** | 0.942 | 27.59 | 0.930 |
| SiMAE | ✓ | ✓ | ✓ | **26.49** | **0.916** | **27.18** | **0.944** | **29.00** | **0.929** | 28.61 | **0.943** | **27.82** | **0.933** |

Table 15: Ablation on subject token (S.T.), information bottleneck (I.B.), and SIS loss on ADNI.

| | S.T. | I.B. | SIS | T1-w PSNR | SSIM | T2-w PSNR | SSIM | PD PSNR | SSIM | Average PSNR | SSIM |
|---|---|---|---|---|---|---|---|---|---|---|---|
| Baseline | | | | 25.87 | 0.796 | 28.03 | 0.840 | 28.21 | 0.881 | 27.37 | 0.839 |
| | ✓ | | | 25.76 | 0.791 | 28.02 | 0.840 | 27.99 | 0.880 | 27.26 | 0.837 |
| | ✓ | ✓ | | 26.22 | 0.805 | 28.32 | 0.849 | 28.47 | 0.885 | 27.67 | 0.846 |
| | ✓ | | ✓ | 26.32 | 0.805 | 28.75 | 0.853 | 29.09 | 0.890 | 28.05 | 0.849 |
| SiMAE | ✓ | ✓ | ✓ | **26.69** | **0.814** | **28.85** | **0.855** | **29.07** | **0.892** | **28.20** | **0.854** |

Table 16: One-stage vs. two-stage training on BraTS. (RTM: Random Token Masking, RCM: Random Contrast Masking)

| Strategy | FLAIR PSNR | SSIM | T1-w PSNR | SSIM | T1Gd PSNR | SSIM | T2-w PSNR | SSIM | Average PSNR | SSIM |
|---|---|---|---|---|---|---|---|---|---|---|
| RTM | 24.59 | 0.905 | 26.09 | 0.938 | 28.94 | 0.927 | 28.00 | 0.938 | 26.91 | 0.927 |
| RCM | 25.03 | 0.894 | 25.13 | 0.930 | 27.69 | 0.921 | 27.49 | 0.932 | 26.34 | 0.919 |
| RTM+RCM | 24.44 | 0.890 | 24.88 | 0.928 | 27.18 | 0.916 | 26.97 | 0.927 | 25.87 | 0.915 |
| RTM→RCM | **26.39** | **0.911** | **26.67** | **0.940** | **29.04** | **0.928** | **28.43** | **0.940** | **27.63** | **0.930** |

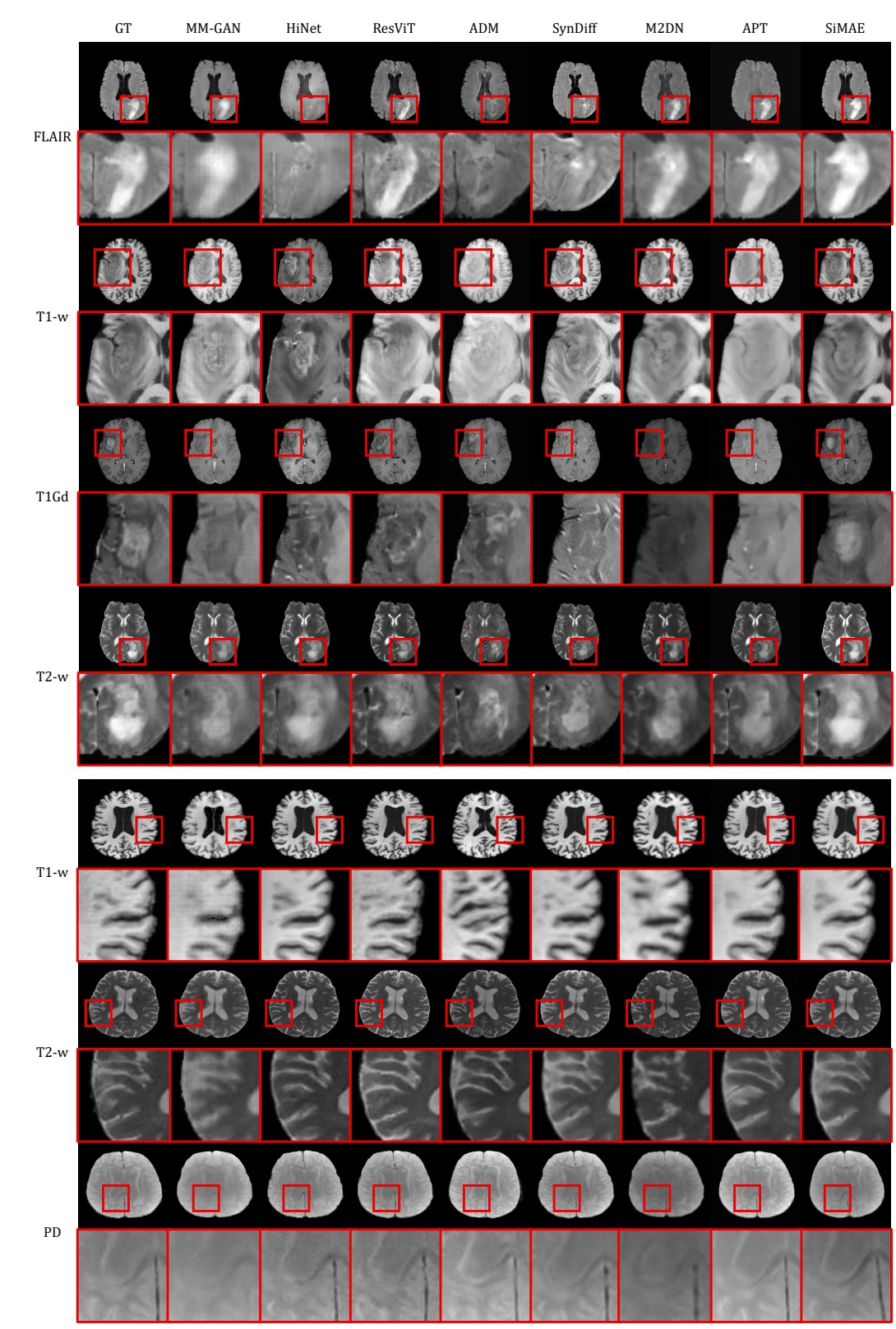

Figure 10: Additional qualitative results on BraTS (top) and ADNI (bottom). Red boxes highlight the main differences with zoomed images.

Table 17: One-stage vs. two-stage training on ADNI. (RTM: Random Token Masking, RCM: Random Contrast Masking)

| | T1-w | | T2-w | | PD | | Average | |
|---|---|---|---|---|---|---|---|---|
| Strategy | PSNR | SSIM | PSNR | SSIM | PSNR | SSIM | PSNR | SSIM |
| RTM | 25.79 | 0.796 | 27.88 | 0.838 | 27.67 | 0.879 | 27.11 | 0.838 |
| RCM | 25.77 | 0.803 | 27.65 | 0.834 | 27.86 | 0.875 | 27.09 | 0.837 |
| RTM+RCM | 24.72 | 0.770 | 26.99 | 0.816 | 27.24 | 0.871 | 26.32 | 0.819 |
| RTM→RCM | **25.87** | **0.796** | **28.03** | **0.840** | **28.21** | **0.881** | **27.37** | **0.839** |

Table 18: Parameters and inference time vs. diffusion-based baseline.

| | APT (Shin et al., 2025) | SiMAE (Ours) |
|---|---|---|
| Total parameters | 200.3M | 213.8M |
| Inference time | 9.06s | 0.06s |

Table 19: Average PSNR and SSIM under different SIS weight $\lambda$ and temperature $\tau$ on BraTS.

| | $\lambda = 0.25$ $\tau = 0.5$ | $\lambda = 0.5$ $\tau = 0.25$ | $\lambda = 0.5$ $\tau = 0.5$ | $\lambda = 0.5$ $\tau = 1.0$ | $\lambda = 1.0$ $\tau = 1.0$ |
|---|---|---|---|---|---|
| PSNR | 27.64 | 27.74 | **27.82** | 27.79 | 27.78 |
| SSIM | 0.932 | 0.932 | **0.933** | 0.932 | 0.932 |

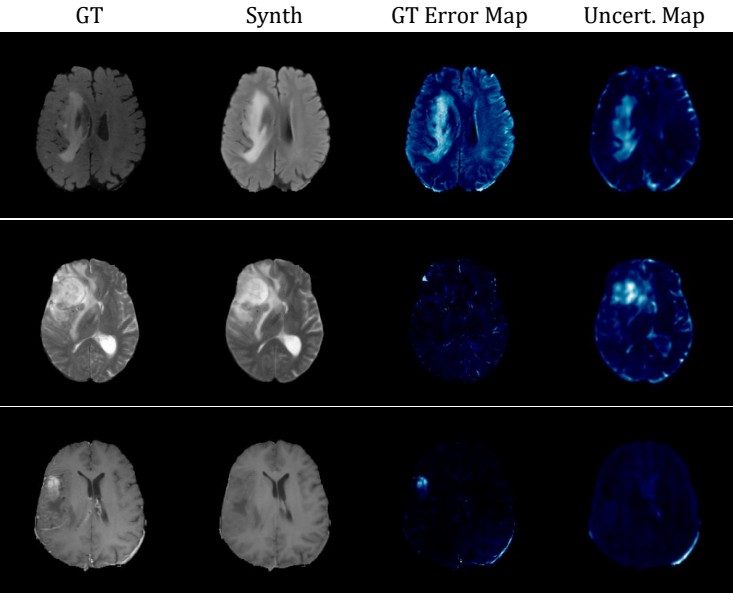

Figure 11: Failure cases. Top: Incorrect target-contrast appearance despite plausible anatomy. Middle: Local synthesis error not captured by the uncertainty map. Bottom: Missed tumor and poor uncertainty estimation.

