# OpenReview forum: "SiMAE: Subject-identity Separation Latent Masked Autoencoder for Multi-contrast MRI Synthesis and Uncertainty Estimation"
_ICLR.cc/2026/Conference — ICLR 2026 Conference Withdrawn Submission_

### Official Review · Reviewer_NGdk · 2025-10-16

**Soundness:** 3
**Presentation:** 3
**Contribution:** 3
**Rating:** 4
**Confidence:** 4

**Summary:**

This paper tackles the task of synthesizing missing MRI modalities using a two-stage latent masked autoencoder (MAE) framework. The method first learns general anatomical context through random token masking and then fine-tunes for cross-contrast synthesis via contrast-level masking. It integrates a subject-identity mechanism and produces uncertainty maps for reliability analysis. The experiments are comprehensive on standard datasets and demonstrate consistently better performance and efficiency compared to existing GAN- and diffusion-based methods.

**Strengths:**

1. The proposed uncertainty map offers intuitive and quantitative interpretability, providing valuable insights into the model’s inference reliability.

2. The experiments are comprehensive, covering multiple datasets and baselines, and consistently demonstrate superior reconstruction quality and efficiency.

3. The two-phase curriculum training strategy (token masking → contrast masking) in the latent-space MAE effectively enhances both general anatomical understanding and cross-modality synthesis performance.

**Weaknesses:**

1.The model processes each modality as independent 2D slices, without explicitly modeling inter-slice spatial continuity. Since MRI data are inherently 3D, incorporating volumetric context or evaluating 3D consistency metrics (e.g., 3D-SSIM) would strengthen the study.

2.The paper lacks a comparison with latent-diffusion baselines. Given that the proposed approach also operates in latent space, such a baseline would clarify whether the performance gains stem from the MAE design or the latent representation itself.

3.While the method claims to handle arbitrary missing-contrast scenarios, it is unclear how it performs in extreme cases where only a single contrast (e.g., T1) is available. Evaluating this setting would reveal the model’s capacity to capture anatomy priors independent of cross-modal cues.

**Questions:**

As mentioned in the above weekness:

1.Have the authors considered modeling inter-slice continuity or reporting 3D metrics (e.g., 3D-SSIM) to evaluate volumetric consistency?

2.It would be helpful if the authors could discuss the absence of a latent-diffusion baseline, which would provide a fairer comparison for latent-space generative approaches.

3.How does the model perform when only a single contrast (e.g., T1) is available as input?

---

### Official Review · Reviewer_j6Uu · 2025-10-23

**Soundness:** 2
**Presentation:** 3
**Contribution:** 2
**Rating:** 4
**Confidence:** 4

**Summary:**

The paper proposes a novel multi-contrast MRI synthesis model named SiMAE. The method builds upon the concept of a latent masked autoencoder, trained to reconstruct missing MRI contrasts from the available ones. A multi-contrast tokenizer with a shared encoder maps each contrast image into a common latent space, from which all contrast images are jointly reconstructed using a decoder that aggregates cross-contrast information. In addition, the authors introduce a subject token and a subject-identity separation loss to better capture anatomical identity and subject-specific features. Experimental results show that SiMAE achieves superior synthesis accuracy compared to existing methods. Furthermore, the framework supports the generation of uncertainty maps through iterative latent masking, highlighting low-confidence regions and enhancing the interpretability of the model.

**Strengths:**

- The paper is generally well written and easy to follow, presenting its motivation and contributions clearly.
- Adapting the latent masked autoencoder paradigm for multi-contrast MRI synthesis is both novel and well-motivated.
- The ability to generate uncertainty maps through iterative latent masking enhances the interpretability of the model, which is particularly important for clinical reliability and decision support.
- The experimental results on the BraTS 2021 and ADNI datasets clearly demonstrate the effectiveness of the proposed model compared to existing approaches considered in the evaluation.
- The model produces high-quality reconstructions that are comparable and even superior to diffusion-based methods, while offering significantly faster inference speeds.
- The proposed Subject-Identity Separation (SIS) loss effectively promotes the disentanglement of anatomical identity and subject-specific features, contributing to more consistent and robust synthesis performance.

**Weaknesses:**

- The authors argue that the proposed method preserves fine anatomical details and pathological structures better than existing models. However, the evaluation relies solely on quantitative metrics computed automatically. To support this claim and strengthen clinical relevance, the paper should include expert validation by radiologists or a reader study assessing anatomical and pathological fidelity.
- Similarly, to assess the diagnostic utility of the synthesized images, downstream tasks such as tumor segmentation could be performed. This would help demonstrate whether the synthesized contrasts preserve diagnostically meaningful information.
- The related work and experimental evaluation sections omit several recent diffusion-based approaches, including D2Diff [1] and MRDiff [2]. These models should be discussed, and comparative experiments should be conducted to ensure a comprehensive and up-to-date evaluation.

     [1] Dayarathna et al., D2Diff : A Dual Domain Diffusion Model for Accurate Multi-Contrast MRI Synthesis, MICCAI 2025

     [2] Shin et al., Physics-Driven Signal Regularization in Diffusion Models for Multi-contrast MR Image Synthesis., MICCAI 2025

**Questions:**

- See the weaknesses listed above.
- Could the computed uncertainty maps be incorporated into the training objective (e.g., uncertainty-weighted reconstruction or regularization) to further improve synthesis fidelity and robustness?

---

### Official Review · Reviewer_FL4a · 2025-10-25

**Soundness:** 2
**Presentation:** 2
**Contribution:** 2
**Rating:** 2
**Confidence:** 4

**Summary:**

This paper presents a method for multi-contrast MRI synthesis using masked autoencoders (MAE). The proposed method features a subject token trained with subject-identity separation (SIS) loss, aimed for distinguishing subject identities in the training data. A latent space framework is adopted to improve processing speed on high-dimensional images. A curriculum learning strategy is employed to train the MAE, which first masks input tokens at arbitrary positions, and then masks all the tokens in a single contrast for missing-contrast synthesis. An uncertainty estimation method is also presented. Experiments were conducted on two brain MRI datasets for synthesis of various contrasts such as T1, T2, PD, and FLAIR.

**Strengths:**

- Comparison was conducted against a substantial number of existing methods. Extensive ablation study is conducted to analyze the contribution of each proposed component.
- A candid and open discussion of limitation and failure cases of the proposed method is provided in Section 6 and Figure 11, demonstrating examples of contrast shift and missing tumor.

**Weaknesses:**

- The novelty of the proposed method is relatively limited. The method appears mainly an application and combination of various existing techniques (MAE, latent diffusion, masked image modeling, contrastive learning, curriculum learning, and uncertainty estimation).
- More intuitions and justifications can be provided for the main novelty: the subject token and SIS loss.
- The proposed SIS loss is an application of the contrastive loss. Sufficient discussion on this similarity should be included in the main body, which is currently missing.
- Empirical results lack notions of statistical significance, e.g., standard deviation, and improvements are marginal for some critical ablations.

**Questions:**

- Can the authors provide more intuition on why capturing subject-identity information benefits missing contrast synthesis? These two tasks seem unrelated. The paper claims that the subject token captures "unique anatomical identities"(Section 3.3). Can the authors clarify what "anatomical identities" mean and how they relate to the synthesis task?

- It is even more confusing that removing the subject token from the decoder yields better results in ablation. The authors interpret this as creating an "information bottleneck". Although the bottleneck is good for representation learning with an autoencoder, it is generally not desired in synthesis, which benefits from networks without bottlenecks like U-Net.

- In the ablation study on subject token (Table 3), one variant implements the subject token without the SIS loss. This is confusing -- without the loss, what supervision is applied to the subject token to constrain its learning? What information will the subject token capture? It is also confusing that slight performance gain was achieved by this variant, and interpreted by the authors as a positive result.

- The proposed SIS loss is an application of the contrastive loss, by treating images from different subjects as negative pairs and not considering positive pairs. Contrastive loss has been used in MAEs [1]. However, discussion on this is missing in the main body.

- The uncertainty estimation method can be better justified. Given a predicted contrast, the method randomly masks it, feeds the masked image into the MAE, and computes the difference between the MAE's reconstructions and the original prediction. This only quantifies the uncertainty in the reconstruction (inpainting) step, not the original contrast synthesis step. Why does it reflect the uncertainty in the model's original output?

- The statistical significance of numerical results is unclear, with no standard deviation provided. Furthermore, improvements appear marginal for main contributions in methodology, e.g., 27.82 vs. 27.63 in PSRN by the introduction of subject-identity components (BraTS in Table 3).


Other questions:
- The SIS loss in Eq. 2 assumes that each scan is from a different subject. However, since the data consists of 2D slices, multiple slices can come from the same subject. How are these cases handled?
- In the evaluation of uncertainty estimation, accuracy w.r.t error mask may not be a good metric, given high class imbalance.

[1] Li et al. Mage: Masked generative encoder to unify representation learning and image synthesis. CVPR 2023.

---

### Official Review · Reviewer_svxA · 2025-10-30

**Soundness:** 3
**Presentation:** 3
**Contribution:** 3
**Rating:** 6
**Confidence:** 4

**Summary:**

The paper proposes SiMAE, a latent-space masked autoencoder for multi-contrast MRI synthesis. The model is capable of synthesizing random missing contrasts from the inputs. The model introduces a subject token regularized by a subject-identity separation (SIS) loss, withheld from the decoder to impose an information bottleneck. SiMAE operates in latent space rather than pixel space and further produces uncertainty maps via iterative latent masking. Experiments on BraTS and ADNI show improved PSNR/SSIM and faster inference than diffusion-based methods.

**Strengths:**

1. The paper is well-written and easy to follow in general.
2. The idea of applying MAE in the latent space improves computational efficiency and avoids pixel-space artifacts.
3. The subject token and SIS loss are intuitive mechanisms to decouple anatomy-specific information.
4. The two-stage curriculum training strategy improves model performance.
5. Extensive experiments demonstrate the effectiveness of the proposed method.

**Weaknesses:**

1. The novelty is somewhat incremental, mainly combining known components (MAE, latent autoencoder, dispersive regularization).
2. While the effectiveness of the information bottleneck is validated in the ablation study, it is somewhat counterintuitive that subject-identity information should help the image synthesis. Further analysis/discussion on this would be helpful.
3. The SIS loss does not consider the similarity between subjects.
4. The evaluation is focused on pixel-level metrics (PSNR/SSIM). Metrics like LPIPS and FID can help evaluate the perceptual quality, which are not included.

**Questions:**

1. While the effectiveness of the information bottleneck is validated in the ablation study, it is somewhat counterintuitive that subject-identity information should help the image synthesis. Further analysis/discussion on this would be helpful.
2. On the ADNI dataset, SiMAE performs much better than the baselines compared to BraTS. Any intuition on why?

---

### Note · Authors · 2025-11-13

I have read and agree with the venue's withdrawal policy on behalf of myself and my co-authors.